# Translation, Adaptation and Validation of the Pandemic Fatigue Scale (PFS) in the Greek Language

**DOI:** 10.3390/healthcare10112118

**Published:** 2022-10-22

**Authors:** Evanthia Asimakopoulou, Panagiotis Paoullis, Antonio Shegani, Alexandros Argyriadis, Agathi Argyriadi, Evridiki Patelarou, Athina Patelarou

**Affiliations:** 1Department of Nursing, School of Health Sciences, Frederick University, Nicosia 1036, Cyprus; 2School of Biomedical Engineering & Imaging Sciences, King’s College London, King’s Health Partners, St Thomas’ Hospital, London SE1 7EH, UK; 3Department of Psychology and Social Sciences, Frederick University, Nicosia 1036, Cyprus; 4Department of Nursing, Faculty of Health Sciences, Hellenic Mediterranean University, 71410 Crete, Greece

**Keywords:** COVID-19, pandemic fatigue, scale, translation, validation, Greek language

## Abstract

The growing fatigue of citizens due to the COVID-19 pandemic has already been addressed and its results are visible and threatens citizen compliance. The aim of this study was to translate and validate the Pandemic Fatigue Scale (PFS) in the Greek language. A cross-sectional study was conducted between October 2021 to March 2022. The translation and cultural adaptation process was developed according to the research protocols among the university student population in Cyprus and tested the psychometric properties of PFS. Three hundred thirty-four subjects participated in the study through a web survey, which included general information and the study process. The internal consistency for the total PFS showed good reliability (six items, a = 0.88). A weak statistically significant positive correlation was found between the PFS and the Greek versions of Generalised Anxiety Disorder Assessment—GAD-7 (r = 0.1.96; *p* < 0.001) and the PFS and Patient Health Questionnaire—PHQ-9 (r = 0.173; *p* = 0.002) demonstrating good concurrent validity. Recovering from the pandemic, it is necessary to build systems to detect and respond to future healthcare crises. The results suggest that the psychometric properties of the Greek PFS are satisfactory. The measure of pandemic fatigue allows for identifying fatigue groups for targeted interventions and testing how pandemic fatigue might be reduced in such situations.

## 1. Introduction

To date, there have been millions of diseased and deceased during the coronavirus disease (COVID-19) pandemic and the impact of the pandemic is more than a healthcare problem as it has affected society and the economy with significant burdens globally [1,2,3]. Therefore, during the management of and at the end of the COVID-19 pandemic, physical and psychosocial consequences were studied, such as fear, anxiety, depression, and fatigue [2,4,5,6]. Historically, in similar pandemics, such as the Spanish influenza pandemic, there was a mild first and second wave and a catastrophic third and fourth wave due to the fatigue of the citizens and the pressure exerted on the politicians to return to the normal standards of living [7]. 

It should be noted that in early October 2020, the World Health Organization (WHO) called on European governments to address the growing fatigue of citizens due to the pandemic, estimating that in some cases, it reached 60% [8]. Research worldwide resulted in people complying much better with personal hygiene measures, such as frequent hand washing and mask use, than with measures requiring social contact restrictions [9]. As is well known, university students are a group vulnerable because of their lifestyle, involving social activities within the university campus and the time spent with fellow students [10,11]. 

Fatigue is a concept that has been studied on many different levels in recent years, as a symptom in patients with COVID-19 [12,13], but also as an indication of healthcare professionals, both due to workload [14] and as compassion fatigue [15]. It is also a nursing diagnosis that may indicate an actual or potential condition is a “risk for fatigue”. Fatigue is a factor that should be assessed as it is associated with a variety of symptoms and affects behaviours. Nurses are responsible for the prevention, early detection, and treatment, whether it is the general population or a group of patients [16].

Pandemic fatigue (PF), as named by the WHO, is a reality, and its results are visible and threatens citizen compliance. The WHO defines it as a “lack of motivation to follow the suggested behaviors to protect oneself, which appear gradually over time and are influenced by a series of emotions, experiences and perceptions” [8]. Along with fatigue, depression and anxiety symptoms are common mental health problems in the general population related to pandemics [6]. The latter has also been observed in chronically ill or victims of natural disasters, when there is no clear timetable and the person feels there is no purpose [17,18]. Furthermore, according to studies, media and information exposure during critical events can cause other psychological disorders and fatigue [19,20].

The idea of measuring PF allows the identification of fatigue groups for targeted interventions not only during the COVID-19 pandemic but also in future pandemics. The Pandemic Fatigue Scale (PFS) [4] was the first scale used to investigate who experiences PF and identify related emotions and perceptions in 13 cross-sectional, nationally representative surveys conducted in Denmark and Germany (overall n = 12,191), and has already been translated and validated into the Korean language [10]. The intercultural adaptation of a questionnaire in different languages is crucial, as it allows its use in different countries, comparing countries and conducting meta-analyses. Therefore, this study aimed to translate and validate the Greek version of the PFS and further explore its psychometric properties. 

## 2. Materials and Methods

A cross-sectional study was conducted between October 2021 to March 2022. The translation and cultural adaptation process was developed according to the guidelines used by the World Health Organization (forward and backward translation method). [21,22]. Two healthcare professionals fluent in English performed the forward translation, and the Greek version of the questionnaire was back-translated by one bilingual healthcare professional and one fluent in English. Three experts compared the versions with the original scale to evaluate similarity in meaning and general clarity. Five healthcare professionals and 20 individuals were asked to reflect upon their understanding of each item using a dual response choice (clear or not clear). The inter-rater agreement was 100% and no modifications were applied. The final translated version in Greek was confirmed in discussion with all those involved and then was administered to a larger sample to test its psychometric impact. Two other validated scales were also administered: Generalised Anxiety Disorder Assessment (GAD-7) [23] and Patient Health Questionnaire (PHQ-9) [24], to correlate with the Greek version of the PFS.

### 2.1. Study Population/Sample

In the current study, 334 out of 42,519 [25], the total number of students attending the universities of Cyprus, participated in the web survey, which included general information and the study process. The students came from five, of which all have health sciences schools, while three have medical schools, out of ten universities operating during the academic year 2020–2021. The number of the sample satisfied the minimum number required for factorial analysis, which according to Hatcher and O’Rourke [26], should be the larger of 5 times the number of variables (30) or 100, while according to Nunnally [27], ten times as many subjects as variables (60). From October 2021 to March 2022, Cyprus, like most countries, was going through one of the most severe waves of the COVID-19 pandemic. Hence, all universities adapted to the pandemic restrictions by conducting online courses.

### 2.2. Ethical Considerations

During the study, ethical considerations were followed. The participation was voluntary and anonymous, and informed consent was obtained. Approval was received from Universities’ ethical committees and the developer of the PFS instrument. 

### 2.3. Survey Instruments

#### 2.3.1. Pandemic Fatigue Scale (PFS)

The Pandemic Fatigue Scale (PFS) is a short, valid, and economic measure that may be used in future research and the country-wide monitoring of public opinion during the COVID-19 pandemic as well as from the beginning of future pandemics [4]. The PFS consists of 6 items grouped in two distinct yet highly correlated factors—behavioural and information fatigue—which both add to people’s overall experience of pandemic fatigue. It is a self-report measure rated on a 7-point Likert-type scale (1 = strongly disagree to 7 = strongly agree). The sum of the response scores for all six items provides a measure of participants’ overall pandemic fatigue. The higher the overall PFS score, the greater the pandemic fatigue reported by participants [10]. 

#### 2.3.2. Generalized Anxiety Disorder Assessment (GAD-7)

The Greek version of the GAD-7 scale [23] consists of seven items and assesses the severity of the generalized anxiety disorder. It is a self-assessment tool rated by a 4-point Likert-type scale (0 = not at all to 3 = nearly every day). The total score ranges from 0 to 21, with higher scores indicating greater severity. In the current study, the reliability coefficient of the scale was found to be 0.913.

#### 2.3.3. Patient Health Questionnaire (PHQ-9)

The PHQ-9 scale [24] is a depression scale translated into Greek by Hyphantis et al. [28]. The scale consists of nine items and is a useful tool for detecting major depression and depressive symptoms in the general population. It is a self-assessment tool rated by a 4-point Likert-type scale (0 = not at all to 3 = nearly every day). The total score ranges from 0 to 27, with higher scores indicating greater severity [29]. The PHQ-9 depression severity score is 15. In the current study, the reliability coefficient of the scale was 0.896.

### 2.4. Data Analysis

Descriptive statistics for continuous variables included the mean and standard deviation (mean, ± SD), whereas for categorical variables, frequency counts and percentages [n, (%)] were calculated. Internal consistency of the scale was evaluated using Cronbach’s alpha coefficient, with alpha >0.7 indicating good internal reliability [30]. 

To assess the factorial structure of the Greek version of the PFS, both exploratory factor analysis (EFA) and confirmatory factor analysis (CFA) were performed. The dataset (n = 334) was randomly divided in two parts to obtain two mutually independent samples for the EFA and CFA. First, EFA was used on the first dataset (n = 167) to examine whether the 6 items used to measure pandemic fatigue could be classified into the same two categories (i.e., behavioural and information fatigue) as in the original scale [4]. A Bartlett’s test of sphericity with *p* < 0.05 and a Kaiser—Meyer—Olkin (KMO) measure of sampling adequacy of 0.6 confirmed the suitability of the dataset for the factor analysis [30]. The principal components analysis with varimax rotation (Kaiser’s normalization) was used as the extraction method, whereas parallel analysis was employed to determine the number of factors [31]. Item factor loadings higher than 0.5 were considered satisfactory [30]. 

Next, CFA with structural equation modelling (SEM) was used on the second dataset (n = 167) to examine whether the data fit the original measurement model of PF and thus verify the factor structure. The maximum likelihood method was used for parameter estimation. Standardized estimates of factor loadings as well as (residual) variances were obtained, and modification indices were examined in order to improve the model fit. Specifically, the model goodness of fit was tested considering the following indices: Chi-square value [32], comparative fit index (CFI) [33], Tucker–Lewis index (TLI) [34], root mean square error of approximation (RMSEA) [35] and standardized root mean square residual (SRMR) [36]. The model was considered to have a good fit with *x^2^*/*df* < 5, a RMSEA < 0.1, a SRMR < 0.05, a CFI, and NFI > 0.90 [30]. Item factor loadings higher than 0.6 were considered satisfactory [30]. 

Differences in PF score means in relation to demographic characteristics were also examined (i.e., gender, educational level, infection with COVID-19, and vaccination against COVID-19). Specifically, two independent groups for each categorical demographic variable were created (i.e., gender: male, female, educational level: undergraduate, postgraduate, infection with COVID-19: yes, no, and vaccination against COVID-19: yes, no). Then, the Student’s independent sample t-test was employed to identify any statistically significant difference in PF score means between these groups. In addition, Pearson correlation coefficient was used to analyse the correlation between the PF score and numerical variables (i.e., age). 

Moreover, in order to assess the validity and the psychometric properties of the Greek version of the PFS, correlations (Pearson’s r) between the Greek version of the PFS and other theoretically related scales (i.e., GAD-7 and PHQ-9) were examined. Statistical significance was accepted at the two-sided 0.05. Descriptive statistics analysis, EFA, and hypothesis testing were performed using IBM SPSS version 25. Parallel analysis and CFA were performed using version R 3.6.2.

## 3. Results

Table 1 presents descriptive statistics n, (%) for the responses of the student sample on the six items of the PF scale. Table 2 summarizes participating students’ (n = 334) demographic characteristics. The majority of participants were female (76.6%). The mean age of the student sample was 28.04 (SD = 9.39), and 51.5% of students were undergraduates. The majority (83.5%) were vaccinated against COVID-19, and 31.7% had experienced infection with COVID-19. 

### 3.1. Exploratory Factor Analysis

Table 3 presents the results of the EFA. As mentioned above, EFA was used on half of the dataset (n = 167). The dataset was considered suitable for EFA since Bartlett’s test of sphericity was found to be significant (*p* < 0.001), and the KMO was found to be satisfactory (0.84) [30]. The principal components analysis with varimax rotation (Kaiser’s normalization) was used as the extraction method. In addition, parallel analysis techniques were used to determine the number of factors (Figure 1). In line with the theoretical background and previous empirical findings, the best result was the two-factor solution. This finding is in line with the original scale [4]. Factors can be named as information fatigue (IF) (item 1–3, alpha = 0.84) and behavioural fatigue (BF) (item 4–6, alpha = 0.80). The factor loadings for the six-item solution were higher than 0.5 since they ranged between 0.68 and 0.90 and were considered satisfactory [30]. The factor solution explains 75.32% of the total variance (IF explains 59.76% of the variance, and BF explains 15.56%). The overall internal consistency for the total PF scale showed good internal reliability (six items, alpha = 0.86). 

### 3.2. Confirmatory Factor Analysis

Next, CFA with structural equation modelling was performed on the other half of the dataset (n = 167). The maximum likelihood method was employed for parameter estimations. Table 4 provides an overview of the fit indices for different factor solutions of the CFA. Initially, a one-factor model with the six items (Model 1) was considered. Model 1 gave unsatisfactory fit indices (*x^2^/df* = 9.59, RMSEA = 0.22, CFI = 0.88, 201 TLI = 0.80 and SRMR = 0.07). Next, a model (Model 2) with two factors with the original items’ composition (BF 3 items, IF 3 items) was considered. In line with Lilleholt, Zettler, Betsch and Böhm [4], the model was fitted as a second-ordered structural equation model. The overall pandemic fatigue measure was considered as a second-order latent variable model, and behavioural and information fatigue as first-order latent variables. Model 2 showed fit indices that were not all satisfactory (*x^2^/df* = 6.19, RMSEA = 0.17, CFI = 0.94, TLI = 0.88 and SRMR = 0.06). Therefore, in order to improve the model’s fit, modification indices with values (>10) were examined. Modification indices suggested error correlations between items three and five and between items three and six. Model 3 allowed correlation between the error terms and provided a good fit to the data (*x^2^/df* = 2.57, RMSEA = 0.09, CFI = 0.99, TLI = 0.96 and SRMR = 0.03). These covariances of item errors may indicate similar conceptual content since it presents a unique variance origin [37]. The fully standardized factor loadings and (residual) variances for the second-order models 2 and 3 are presented in Figure 2. The factor loadings for the six-item solution were higher than 0.5 for both models (Models 2 and 3) and were considered satisfactory [30].

For the chosen model (Model 3), the factor scores for the second-order latent variable pandemic fatigue (PFS) were computed for the whole sample (n = 334). Table 2 summarizes participating students’ (n = 334) PFS mean scores and demographic characteristic. Undergraduate students experienced significantly less fatigue (MPFS = 3.52; SD = 1.47), than postgraduate students (MPFS = 3.99; SD = 1.32) with statistical significance (t332 = −3.084 and *p* = 0.002). In addition, students infected with COVID-19 reported a significantly higher-level PF score (MPFS = 4.22; SD = 1.37) compared with students that had not been infected (MPFS = 3.53; SD = 1.38), with statistical significance (t332 = −4.229 and *p* < 0.001). Vaccinated students reported significantly lower level of fatigue (MPFS = 3.61; SD = 1.39), compared with non-vaccinated students (MPFS = 4.49; SD = 1.32), with statistical significance (t332 = 4.320 and *p* < 0.001). No significant differences in the PF score were identified when age and gender were examined. 

### 3.3. Concurrent Validity

Concurrent validity and the psychometric properties of the Greek version of the PFS were examined by computing the correlations (Pearson’s r) between the Greek version of the PFS (Model 3, n = 334), the self-reported anxiety score (GAD-7) and the depression score (PHQ-9). Means ± SDs of the GAD-7 and PHQ-9 scores were found to be 7.33 ± 5.55 and 7.75 ± 6.32, respectively. A weak statistically significant positive correlation was identified between the PFS and GAD-7 (r = 0.196; *p* < 0.001) and between the PFS and PHQ-9 (r = 0.173; *p* = 0.002) demonstrating a good concurrent validity of the PFS.

## 4. Discussion

Pandemic fatigue is a contemporary concept that refers to mental and physical exhaustion due to the COVID-19-related restriction. This is a typical and expected response to a prolonged public health crisis—mainly due to the chronic restrictions to mitigate the new SARS-CoV-2 virus that has affected everyone’s daily life, even those who have not been infected [8,38].

This study aimed to translate and explore the psychometric properties of the PFS in the Greek version among university students in Cyprus. The two factors named information fatigue (IF) and behavioural fatigue (BF), which resulted from the original scale [4], are also confirmed by the results of this study.

The first factor (information fatigue) includes the items: “I am tired of all the COVID-19 discussions in TV shows, newspapers, and radio programs, etc., “I am sick of hearing about COVID-19” and “When friends or family members talk about COVID-19, I try to change the subject because I do not want to talk about it anymore”. The overwhelming amount of available information on COVID-19 and the doubt about which sources are trustworthy have led to pandemic fatigue [20,39]. So-called “infodemic” has correlated with weakened mitigation measures [20,40]. In a study about COVID-19 message fatigue, most participants reported being tired of constantly hearing about COVID-19, with the reminder to wear a mask being more tedious [41].

The second factor (behavioural fatigue) with the items: “I am tired of restraining myself from saving those who are most vulnerable to COVID-19”, “I am losing my spirit to fight against COVID-19”, and “I feel strained from following all of the behavioural regulations and recommendations around COVID-19” indicates the emerging demotivation to engage in protection behaviours as a result of complacency, alienation, and hopelessness [8,38]. The term “behavioural fatigue” refers to any factor negatively affecting compliance with regulations over time. The term could also refer to a physiological mechanism that decreases people’s ability to behave in a certain way after a prolonged time, analogous to muscular fatigue [42]. A similar mechanism of behavioural fatigue is observed in adherence failure to medication, particularly for those with chronic diseases [43].

In our study, the PFS was found to be weakly positively correlated with the GAD-7 (r = 0.196; *p* < 0.001) and PHQ-9 (r = 0.173; *p* < 0.002) demonstrating good concurrent validity. The scales are not measuring exactly the same thing, so the correlation is not strong, but there is a positive correlation indicating a possible sign of comorbidity of pandemic fatigue with anxiety and depression that requires further investigation [6,10].

Further analysis of the study highlighted that students infected with COVID-19 reported a higher level of PF score, probably due to the disease and the length of restrictions they suffered. It has been observed that people respond positively to restrictive measures at the beginning and when they know the period of restrictions [8,9]. Undergraduate students experienced less PF than postgraduate students. People with a higher level of knowledge related to COVID-19 are more likely to find the information they need, develop protective behaviours, and manage distress levels [44,45]. On the other hand, emotions and contextual factors may significantly impact behaviours more than knowledge [20]. In addition, the sample consisted of young people (mean = 28 years) that are more likely to experience PF, perhaps because they feel that they constantly have to put the needs of those at a higher risk (e.g., senior citizens) before their own [46]. Vaccinated students reported a lower level of PF than non-vaccinated students, resulting in total agreement with studies in the general population [47,48]. Regarding age and gender, no significant differences in PF scores were identified in this study. This has also been observed in other studies, as no correlation was found between age and self-reported fatigue, nor an effect of gender [49]. Although women are expected to report generally greater trait levels of fatigue than men due to biological and social factors, when young age is included as a factor, no difference is observed [50].

Our study is not without limitations. To our knowledge, this is the first study of translation and validation of a PFS in the Greek language. Although the reliability and validity of the scale were satisfactory, further studies in different populations and settings should test the scale’s psychometric properties. Moreover, the sample of this study consisted of 334 participants, which is adequate but limited to students. Information and selection bias in the study sample may also have occurred from the procedure used to select subjects and factors that influenced participation in the study. More studies with larger and more representative samples are needed to enhance the scale and provide more information about the PFS.

## 5. Conclusions

PFS is a short, valid, and economic measure that could be used in future research and the country-wide monitoring of public opinion during the COVID-19 pandemic, as well as from the beginning of future pandemics and healthcare crises. The Greek version of the PFS has demonstrated satisfactory reliability, and the factor analysis indicated two different factors. Hence, it is a reliable and valid tool for detecting PF among the general population. The factors in the scale indicate that the actions should be targeted regarding information and behaviour, meaning that a strategy aimed at providing public health information alone may not be the most effective. These results may inform policymakers about who should target pandemic mitigation interventions. More specifically, knowing the risk factors for COVID-19 PF, more appropriate and effective measures could be taken. Nurses can use it in their clinical practice in the community, and future cross-sectional and cohort studies are recommended to implement this scale in hospitalized patients and healthcare professionals.

## Figures and Tables

**Figure 1 healthcare-10-02118-f001:**
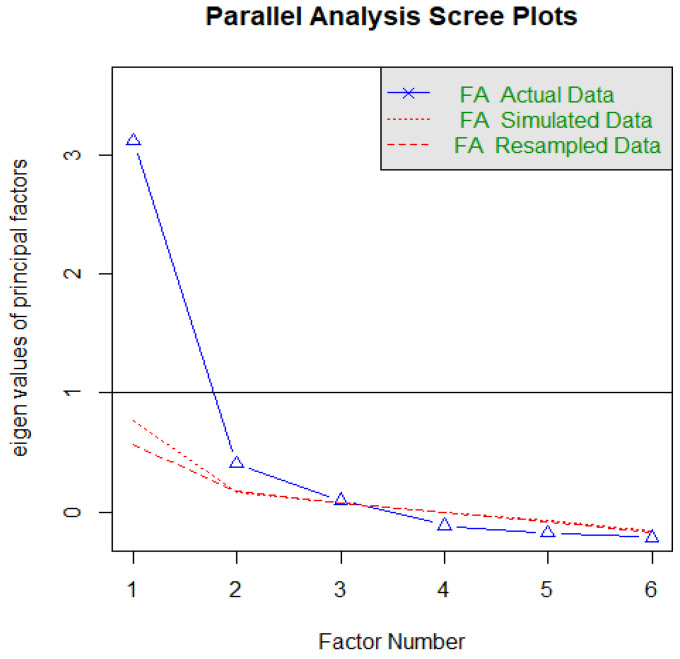
Results of the parallel analysis.

**Figure 2 healthcare-10-02118-f002:**
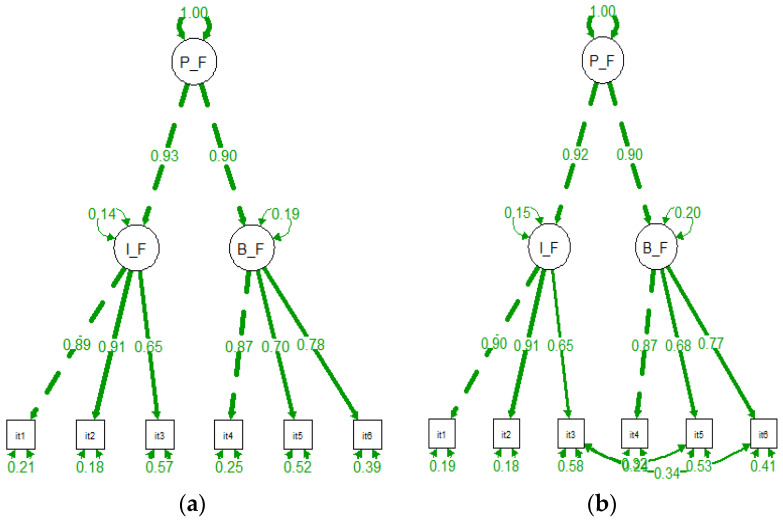
Results of the confirmatory factor analysis: (**a**) Model 2 (6-items model (2 factors) (**b**) Model 3 (6-item model (2 factors, correlated errors).

**Table 1 healthcare-10-02118-t001:** Responses to the pandemic fatigue scale, n (%).

	Strongly Disagree	Disagree	Somewhat Disagree	Neutral/Neither Disagree nor Agree	Somewhat Agree	Agree	Strongly Agree
I am tired of all the COVID-19 discussions in TV shows, newspapers, radio programs, etc. (item 1)	25(7.5)	27(8.1)	15(4.5)	27(8.1)	51(15.3)	59(17.7)	130(38.9)
I feel strained from following all of the behavioural regulations and recommendations around COVID-19. (item 4)	49(14.7)	67(20.1)	19(5.7)	51(15.3)	51(15.3)	52(15.6)	45(13.5)
I am sick of hearing about COVID-19. (item 2)	31(9.3)	32(9.6)	18(5.4)	47(14.1)	48(14.4)	52(15.6)	106(31.7)
I am tired of restraining myself to save those who are most vulnerable to COVID-19. (item 5)	101(30.2)	71(21.3)	20(6)	43(12.9)	30(9)	33(9.9)	36(10.8)
When friends or family members talk about COVID-19, I try to change the subject because I do not want to talk about it anymore. (item 3)	56(16.8)	66(19.8)	32(9.6)	53(15.9)	33(9.9)	43(12.9)	51(15.3)
I am losing my spirit to fight against COVID-19. (item 6)	93(27.8)	70(21)	24(7.2)	52(15.6)	26(7.8)	32(9.6)	37(11.1)

**Table 2 healthcare-10-02118-t002:** Demographic characteristics of students (n = 334), n (%), PFS (6 items) means, and standard deviations (model with 6 items).

		PFS		
		*n* (%) or Mean *±* SD	Mean *±* SD	Test
**Gender**	Male	78(23.4)	3.80 *±* 1.41	t_332_ = 0.327, *p* = 0.744
	Female	256(76.6)	3.74 *±* 1.42	
**Age**		28.04 *±* 9.39		*r* = 0.069, *p* = 0.206
**Level of education**	Undergraduate	172(51.5)	3.52 *±* 1.47	t_332_ = −3.084, *p* = 0.002
	Postgraduate	162(48.5)	3.99 *±* 1.32	
**Infected with COVID-19**	yes	106(31.7)	4.22 *±* 1.37	t_332_ = −4.229, *p* < 0.001
	no	228(68.3)	3.53 *±* 1.38	
**Vaccinated against COVID-19**	yes	279(83.5)	3.61 *±* 1.39	t_332_ = 4.320, *p* < 0.001
	no	55(16.5)	4.49 *±* 1.32	

**Table 3 healthcare-10-02118-t003:** Explanatory factor analysis (principal component analysis extraction; varimax; Kaiser’s normalization) (n = 167).

	Model (6 items)	
	Factors	
Item	IF *	BF *
I am tired of all the COVID-19 discussions in TV shows, newspapers, radio programs, etc. (item 1)	**0.857**	0.241
I am sick of hearing about COVID-19. (item 2)	**0.900**	0.155
When friends or family members talk about COVID-19, I try to change the subject because I do not want to talk about it anymore. (item 3)	**0.675**	0.482
I feel strained from following all of the behavioural regulations and recommendations around COVID-19. (item 4)	0.465	**0.678**
I am tired of restraining myself to save those who are most vulnerable to COVID-19. (item 5)	0.09	**0.876**
I am losing my spirit to fight against COVID-19. (item 6)	0.321	**0.806**
% of variance	59.76	15.56
**Cronbach’s alpha**	**0.84**	**0.80**
**Cronbach’s alpha overall**	**0.86**	
	KMO = 0.81, Bartlett’s Test of Sphericity = 446.47 (*p* < 0.001)	

* Information fatigue: IF and behavioural fatigue: BF.

**Table 4 healthcare-10-02118-t004:** Results of the confirmatory factor analysis: estimated models (n = 167).

Model		*χ^2^*	*df*	*χ^2^/df*	RMSEA	CFI	TLI	SRMS
**Model 1:**	6 items (one factor)	86.33	9	9.59	0.22	0.88	0.80	0.07
**Model 2:**	6-item model (two factors)	49.48	8	6.19	0.17	0.94	0.88	0.06
**Model 3:**	6-item model (two factors, correlated error– item 3 <-> item 5 and item 3 <-> item 6)	15.42	6	2.57	0.09	0.99	0.96	0.03

## Data Availability

Not applicable.

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
