# Peer review of "Translation, Adaptation and Validation of the Pandemic Fatigue Scale (PFS) in the Greek Language"

_healthcare, 2022, doi:10.3390/healthcare10112118_

Round 1

Reviewer 1 Report

Comments are described within the attached PDF  in the comment section.

Author Response

Dear Sir or Madam,

The manuscript has been revised according to your comments and a new file has been uploaded.

Regarding the data analysis, based on our new approach of splitting the sample and doing the analysis (EFA, CFA) on two samples, as your instructions, the results have been changed and are re-presented both in the text and in the Tables.

To support our data and results thirteen (13) new References were added.

Thank you for the very accurate and useful comments which we hope have structurally and conceptually improved our manuscript.

We look forward to you accepting our article for publication in Healthcare.

Reviewer 2 Report

In my opinion the paper was informative and will provide a valuable source document for anyone requiring a primer to know and understand this issue. Some changes are needed:    

  • Line 23: In it not stated in the Abstract, and it must be stated, what pandemic this refers to. 
  • Line 31:  Define the abbreviation GAD-7. 
  • Line 32:  Define the abbreviation PHQ-9.  
  • Lines 40-41; When writing about the `the impact of the pandemic` it must be said precisely what pandemic it is about. Also, in the same sentence it is important to emphasize the impact and consequences of the pandemic, because there are millions of diseased and deceased during the pandemic, then there are the economic effects, etc. Also, it is good to use data from the World Health Organization when referring to a pandemic of a disease.  
  • Line 42: Define the abbreviation COVID-19. 
  • Line 47: Define the abbreviation WHO. 
  • Lines 51-53: These 2 sentences should be deleted or for each of them an appropriate reference must be cited which will confirm the stated claims. 
  • Lines 53-54: Be precise whether in students, in the given context, it is about the lifestyle or professional/academic obligations.   
  • Lines 73-77: Is the ref No 9 the appropriate and correct reference for the claim in this sentence. 
  • Line 93: Cite the appropriate reference for GAD-7. 
  • Line 93:  Cite the appropriate reference for PHQ-9. 
  • Lines 95-100: Reconstruct the section Materials and Methods, so that instead of the subsection `Sample Data collection` you inscribe a subsection: `Study Population/Sample` (with a description of how many universities were included in the study, how many students in total are there at those universities, how many medical faculties and how many faculties of health sciences). Because this cross-sectional study was conducted between October 2021 and March 2022, when most of the countries were going through one of the most severe waves of the COVID-19 pandemic, it is necessary to describe how the studies and lectures were organized at the universities included in this study.
  • Lines 105-108: Instead of the subsection `Data analysis` (since the end of the section Materials and Methods already has the subsection `Statistical Analysis`), inscribe `Data Collection`. Within the new section `Data Collection`, describe how participants were recruited for this study. 
  • Lines 109-124: Since this part of the text lists the three used scales, it is necessary to reorder the numbers of all three subsections in an appropriate manner. 
  • Lines 123-124: Inscribe the appropriate reference which has published the mentioned results. 
  • Line 277-289: Discuss the absence of differences between males and females. 
  • Lines 290-296: Discuss informative bias in the study sample in this study as a limitation of the study. 
  • Lines 290-296: Discuss the selection bias in the study sample in this study as a limitation of the study.   

Author Response

Dear Sir or Madam,

The manuscript has been revised according to your comments and a new file has been uploaded.

Thank you for the very accurate and useful comments which we hope have structurally and conceptually improved our manuscript.

We look forward to you accepting our article for publication in Healthcare.

Round 2

Reviewer 1 Report

Manuscript has been improved.

Reviewer 2 Report

Thank you to the authors for their responses to my comments. The authors have addressed all of the issues highlighted in my review. The revised manuscript is clear, readable and informative and will provide a valuable findings for this issue.